# Developing an evaluation approach for the in-depth review of a new undergraduate medical programme as a complex system

Detlef R. Prozesky ⬡ *◔, Masego B. Kebaetse ⬡◔, Mpho S. Mogodi◔, Mmoloki C. Molwantwa ⬡◔

Department of Medical Education, Faculty of Medicine, University of Botswana, Gaborone, Botswana

◔ These authors contributed equally to this work.
* prozeskyd@ub.ac.bw

## Abstract

### Background

A new undergraduate medical programme was instituted at the University of Botswana in 2009. In 2016, the Faculty of Medicine decided to conduct a comprehensive review of the programme. Participants at a planning workshop decided the review had to lead to an in-depth understanding of the programme. The challenge was to develop an evaluation process to achieve this aim.

### Approach

Standards and theories used in other programme evaluations were investigated. A steering committee developed a 'six-step process' for the review of the programme as a complex system. The process used three evaluation models and various sources of evaluation standards to derive 90 evaluation questions. Quantitative and qualitative data were collected using 96 different instruments, with data triangulation prominently featured. Data analysis used an interpretivist approach. The review process was validated against the Cilliers's ten features of a complex system.

### Validation

The review process was validated against Cilliers's ten features of a complex system. We found that the 'six-step process' illuminated each of these features in the MBBS programme in turn, and was therefore a valid way of evaluating this programme as a complex system.

### Discussion

In the process of designing and using the 'six-step process' to evaluate a complex medical programme important lessons were learnt: starting the process with complexity theory at the forefront; being as inclusive as possible in data collection; also applying complexity theory to the evaluation of smaller programme components as 'mini-complex systems'; and managing one's inevitable insider bias.

**Data Availability Statement:** In this case there are restrictions in place that prevent the public sharing of our minimal data. The consent forms we used asked study participants to agree to 'provide the

information requested, which will be incorporated (anonymously) in the Review Report only'. We are therefore not able to make these data available without restrictions. Readers which wish to have access to the data supporting this research must please apply to: The Office for Research and Development, University of Botswana (research. ethics@ub.ac.bw).

**Funding:** The author(s) received no specific funding for this work.

**Competing interests:** The authors have declared that no competing interests exist.

## Conclusion and utility

The 'six-step process' as it stands or in adaptation is likely to be useful in similar situations, where evaluators perceive the object of their evaluation to be a complex system, or a component of such a system.

## Background

Programme evaluation, the process of collecting and analysing information about the design, implementation, and outcomes of a programme [1], is an essential element of quality assurance of educational programmes. Several reasons for conducting evaluations include ensuring teaching is meeting students' learning needs, identifying areas where teaching can be improved, informing the allocation of faculty resources, providing feedback and encouragement for teachers, identifying and articulating what is valued by medical schools, and (significantly) facilitating the development of the curriculum [2]. Evaluations occur at both the undergraduate [3–6] and postgraduate [7, 8] levels; on established programmes [9, 10], recently introduced ones [4] or recently reformed ones [11, 12]; may be comprehensive [4, 13] or focus on particular elements of a programme [3, 5, 14, 15]; and may use cross-sectional [9, 16] (sometimes with the stated intention of repeats) [11, 17], cyclical [13, 18] or longitudinal [4, 10] designs. Various data collection methods include student evaluation surveys [3, 14], questionnaires [11, 13], focus group discussions [16, 18], observations [15], and document analysis [5, 10]. Often, several methods are used together to triangulate data collected to arrive at conclusions [3, 6, 9]. Evaluators sometimes use validated instruments [12, 17] but often design instruments fit-for-purpose [14, 16], which are sometimes validated before use [11].

Implicit in evaluation is that decisions are made about the quality or value of a programme, which requires identifying standards against which it is to be measured [19]. In some cases, national standards are used [10, 12, 17] or international standards such as those of the World Federation for Medical Education (WFME) [20], and Social Accountability for Medical Schools [21]. In other cases, the standards seem to be implicit, derived from what is seen to be desirable programme processes and outcomes [3, 9, 18].

Besides the scope, design, data collection methods and standards, theories are also a significant consideration. Generally, group theories underpinning programme evaluation have been classified based on reductionist, systems, and complexity theories [22]. The reductionist theory posits that programmes are static and closed. Reducing any programme to its obvious components and measuring them makes it possible to predict how their changes will affect the outcome. This perspective has been critiqued as leading to 'straightforward cause-and-effect models, linear predictability, and a reductionist, atomistic, analytically fragmented approach to understanding phenomena' [23]. Models and frameworks underpinned by reductionist theory include Kirkpatrick's four-level evaluation model, Bloom's taxonomy [24], and 'task-oriented evaluation' [7].

On the other hand, systems theory postulates that there is more to a system (such as a programme) than its component parts, and the relationships between the elements and their contributions to the system may be rationally explained [25]. As such, the 'closed' system modelled by reductionist theory 'has little explanatory power when it comes to large, open systems where elements interact with each other and their environments through communication and feedback' [25]. Such systems may merely be 'complicated' if their many elements can all

be described completely [26]. Models and frameworks underpinned by systems theory include the logic group of models [22].

Building on the systems theory, the advent of the naturalistic paradigm of programme evaluation [27] heralded a shift to models informed by the complexity theory. Complexity theory challenges educators to think of their world and its happenings in terms of processes rather than 'things' [28]. It is based on the understanding that the relationships between components in a system are non-linear, so systems cannot be fully understood by simply analysing their components. These systems are not in equilibrium because they are open to their surroundings and made up of dynamic, unpredictable interactions between their constituent parts. Instead, they adapt to changing situations and develop a history over time [4, 23, 29]. They are capable of self-transformation and able to remain stable despite disturbances [28]. They illustrate a complicated realm at the 'edge of chaos' between deterministic order and chance [30]. Examples of models and frameworks underpinned by complexity theory include the Jorm and Roberts model [31], the Stufflebeam CIPP (context, input, process, output) model [32], and Cilliers's ten features of complex systems [29].

By their nature, medical education programmes are examples of complex systems: dynamic, composed of interacting parts, rarely in equilibrium, non-linear in structure, and affected by many factors, both internal and external to them [22, 28]. These features suggest that the use of the systems theory spectrum of programme evaluations is potentially valuable for comprehensive reviews such as the one we undertook. Thompson et al. reported the increasing popularity of complexity theory in health services research in a 2016 scoping review [33]. In this review, of the 44 studies, 27 were qualitative and 17 were quantitative and mixed methods studies, with case studies as the most common study design. The reported applications of the theory included making conceptual frameworks and planning data analysis and interpretation [33]. However, none of the studies were comprehensive programme evaluations. Furthermore, it has also been found that the type of complexity theory is often only implicit, and the complexity orientation used is much more commonly located using secondary sources rather than primary sources [34].

In this article, we describe a 'six-step process' for the comprehensive evaluation of an undergraduate medical programme as a complex system. We then reflect on our process using Cilliers's ten features of complex systems [29], which have been identified as a comprehensive framework to guide the evaluation of a complex system. Finally, we discuss the lessons learned and best practices for using the 'six-step process' in the comprehensive evaluation of undergraduate medical programmes as complex systems.

## Our programme evaluation approach

### The context

The University of Botswana Faculty of Medicine (UB FoM) offers a five-year, hybrid PBL undergraduate Bachelor of Medicine, Bachelor of Surgery (MBBS) curriculum [35]. The medical school was established through a presidential directive when the existing strategy of training doctors internationally failed to produce the needed numbers for the health system. The first group of 36 undergraduate medical students was admitted in 2009, and the enrolment has been slowly growing to the current 80. The programme has two phases: two years of systems-based basic and pathology science blocks with early clinical exposure (Phase 1) and three years of clinical and community rotations (Phase 2). In all five years, the principal teaching/learning strategies are problem-based and case-based learning.

According to the UB policy, all new programmes must be reviewed after producing their first graduates. In June 2016 the Faculty of Medicine (FoM) started the review of its

undergraduate curriculum with a day-long planning workshop facilitated by the Department of Medical Education. Faculty management, heads of departments, course coordinators and relevant representatives from the University took part in the workshop. The participants made decisions regarding the aims of the review, and evaluation models and standards. A steering committee drawn from those who attended the workshop was elected. It was tasked with implementation of the review, and terms of reference were drawn up for it. The authors took part of the planning workshop and were also members of the steering committee.

## The programme evaluation model

The challenge facing us was developing and using a model that could rationally evaluate the MBBS programme as a complex system. The design of the evaluation model took cognisance of available options in two key issues in educational programme evaluation: the standards against which to make evaluation decisions and the models, theories, and paradigms (explicit or implicit) upon which to base the evaluation. The result was a model in six steps: (1) setting out the aim, (2) considering evaluation models, (3) deciding on evaluation standards, (4) determining research questions, (5) determining methods and carrying out data collection, and (6) data analysis and presentation (Fig 1).

**Step 1: Setting out the aim.** At the planning workshop the team planning the review unanimously decided that we needed to understand the MBBS programme in its 'wholeness'

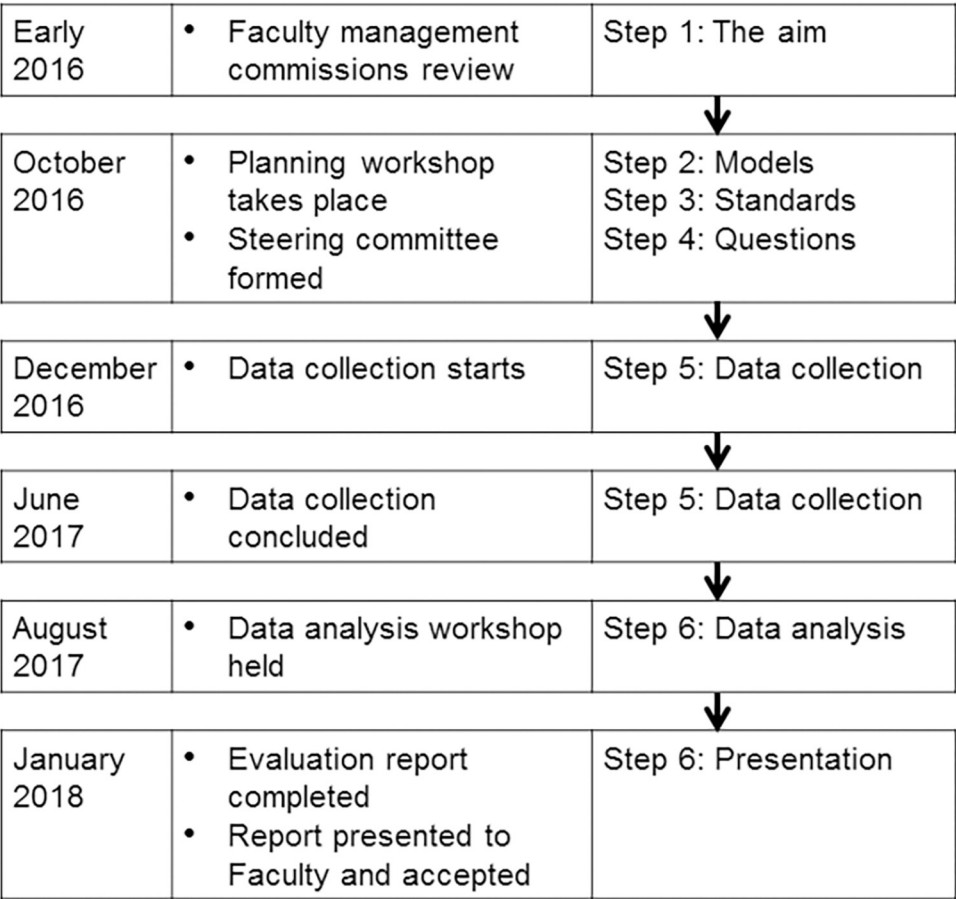

**Fig 1. Diagram of the timeline and major steps of the programme review.**

[25]. The planning team realised that it was a complex, evolving entity that needed to be grasped comprehensively in order to identify strengths and shortcomings confidently, and to plan sensible improvements accordingly. As explained above, this aim directed how we conceived and executed our review.

**Step 2: Considering evaluation models.** We understood that we were evaluating a 'complex system', even if we did not call it that initially. In planning the overall approach to the review, we considered three evaluation models: Stufflebeam's Context, Input, Process and Product (CIPP) model [32], Kirkpatrick's four-level model [36], and McNeil's 'outcomes-based' model (an example of the 'Logic' group of models) [24]. These models were presented and discussed at the planning workshop. In view of the stated aim of a comprehensive review of the programme, it was decided by joint agreement to use the models eclectically. This resulted in an approach at six levels (Table 1):

In further description of the review methods, examples from Level 2 will be used.

**Step 3: Deciding on evaluation standards.** Unless they are purely descriptive, evaluations implicitly require standards to be defined against which programmes are to be judged. The planning workshop discussed various local, national and international sources of standards which participants had previously identified. Local standards included 'Vision, Mission and Values' statements of the University and the Faculty of Medicine; workshop members were also canvassed for issues they felt needed attention. National standards included the revised Botswana National Health Policy [37] and the Botswana Health Professions Council 'Competencies of medical graduates' [38]. International standards to be consulted were Global Standards for Basic Medical Education (WFME) [39], the Global Consensus for Social Accountability of Medical Schools [40], the Lancet Commission report: Health Professionals for a New Century [41], and Core Competencies for Undergraduate Students in Clinical Associate, Dentistry, and Medical Teaching and Learning Programmes in South Africa (The Health Professions Council of South Africa) [42].

In keeping with the previous decision at Step 1 to attempt to understand the MBBS programmes in its 'wholeness' [26] workshop members agreed to use all the standards mentioned above. At subsequent meetings, steering committee members studied each document in depth to elicit standards strongly relevant to the MBBS programme–in other words, the 'must knows'–and then allocated each standard to one of the six levels of the review [S1 and S2 Texts].

**Step 4: Determining research questions.** Steering committee members compared and amalgamated the standards identified in Step 3 and then translated each standard into one or more evaluation questions. As Tackett, Grant, and Mmari have pointed out concerning the

**Table 1. Six evaluation levels informed by three evaluation models.**

| Level | Name | Meaning | Guiding evaluation model |
|---|---|---|---|
| 1 | Programme context | ▪ Community needs, assets/resources, the political climate | ▪ CIPP |
| 2 | Programme inputs: goals and outcomes | ▪ Mission, goals, strategy, plans and policies of the programme<br>▪ Planned programme outcomes | ▪ CIPP<br>▪ McNeil |
| 3 | Programme inputs: resources | ▪ Inputs: programme resources | ▪ CIPP |
| 4 | Programme process: implementation | ▪ Quality of programme implementation<br>▪ Participant reaction to the programme<br>Activities intended to produce outcomes | ▪ CIPP<br>▪ Kirkpatrick<br>▪ McNeil |
| 5 | Programme outputs: direct products | ▪ Short-term outcomes<br>▪ Changes in knowledge, skills, attitudes, competencies<br>▪ Achievement of intended results; impact on learners | ▪ CIPP<br>▪ Kirkpatrick<br>▪ McNeil |
| 6 | Programme outcomes: effects on the target population | ▪ Long-term outcomes<br>▪ How students apply learning<br>▪ Impact on society: changes, benefits | ▪ CIPP<br>▪ Kirkpatrick<br>▪ McNeil |

Table 2. Selected standards and accompanying questions for Level 2 of the MBBS review.

| Evaluation standards | Questions derived from standards (with numbers) | Instruments used to collect data |
|---|---|---|
| The programme must:<br>▪ Define the curriculum model<br>▪ Foster outcomes-based education | 2.1 Is the MBBS programme/ curriculum model defined? Is it outcomes-based and competency-based? | Document analysis:<br>▪ Original MBBS programme<br>▪ Detailed curriculum documents |
| The programme must prevent overload by focusing on conditions relevant to Botswana | 2.4 Are steps taken to prevent content overload? Is the material that is presented relevant to medical practice in Botswana? | Document analysis:<br>▪ Detailed curriculum documents<br>▪ Morbidity/ mortality data for Botswana<br>Interview:<br>▪ Senior academics<br>▪ Block and rotation coordinators |
| The programme must ensure that:<br>▪ Students acquire sufficient knowledge and clinical and professional skills<br>▪ Students spend a reasonable part of the programme in planned contact with patients<br>▪ The amount of time spent in training in major clinical disciplines is specified<br>▪ Appropriate attention is given to patient safety | 2.6 Is skills training/ patient contact structured rationally from Year 1 until the end? Is there sufficient patient contact, is there specified time in major clinical disciplines? Is there attention to patient safety? | Document analysis<br>▪ Detailed curriculum documents<br>▪ Current skills training programme in Phase 1<br>Interview:<br>▪ Clinical HoDs<br>Questionnaire:<br>▪ Senior students |

WFME standards, this is no easy task [20]. Table 2 illustrates some examples for Level 2 (Programme inputs: goals and outcomes). At this stage, we gave each review question a number.

In all, we decided on 90 evaluation questions. Several of these had multiple components–e.g. Question 2.6 in Table 2.

**Step 5: Determining methods and carrying out data collection.** As indicated by the nature of the study aim and research questions, we used a case study and concurrent mixed methods design, collecting both quantitative and qualitative data. An in-depth understanding of the programme required us to collect data from all sources appropriate to each question to arrive at conclusions by triangulation. We gave every staff member, other identified programme stakeholders, and a substantial sample of students the opportunity to contribute information. On approximately 205 occasions, human participants (staff, partners and students) were approached for interviews, focus group discussions, questionnaire completion and observation of their teaching practice. This took place over 12 months, beginning in June 2016. Before data collection started, we trained data collectors for interviews, observations, and focus group discussions. Table 3 summarises the data collection instruments used (96 in all) and the frequency of their intended use: interview guides, focus group guides, questionnaires, observation checklists, and document study guides [S3 Text].

Data collection presented challenges. We did not use all the prepared instruments–practical difficulties arose, e.g. collecting busy academics for focus group discussions. We did not collect data from patients and community members in sufficient breadth and depth. Some categories of respondents did not use the opportunities offered to them to participate.

**Step 6: Data analysis and presentation.** Four members of the steering committee met for a two-week retreat during which data analysis was substantially completed. We carried out data analysis in three stages:

**Table 3. Data collection instruments developed and used for all levels of the review.**

| Instrument | Site of application | Number developed | Use of instruments |
|---|---|---|---|
| Interview guide | ▪ Faculty of Medicine<br>▪ University<br>▪ Partners/ community | 11<br>5<br>8 | 57 conducted |
| Questionnaire | ▪ Faculty of Medicine<br>▪ Partners/ community | 8<br>2 | 73 received |
| Focus group discussion guide | ▪ Faculty of Medicine<br>▪ Partners/ community | 9<br>1 | 5 conducted |
| Observation checklist | ▪ Educational activities<br>▪ Sites where training occurs | 8<br>4 | 68 events/ sites observed |
| Document study guide | ▪ General Faculty of Medicine policies<br>▪ MBBS programme<br>▪ University<br>▪ National/international | 8<br>18<br>7<br>7 | 117 studied |

1. We summarised quantitative data and calculated frequencies using Excel.

2. For the next step, we used an interpretivist paradigm, 'viewing the world through the perceptions and experiences of the participants' and 'trying to understand the diverse ways of seeing and experiencing the world through different contexts' [43, 44]. We considered each question as a free-standing unit and brought all the data (qualitative and quantitative) about it together. We familiarised ourselves with the data, identified emerging themes, and coded them under two headings: 'achievements' and 'issues to consider/ areas for improvement/ challenges'.

3. Based strictly on the findings, we made recommendations under each question. An example of how we presented findings for each research question is given in Table 4.

**Table 4. Findings presented for research question 2.5.**

**Are steps taken to prevent content overload? Is the material that is presented relevant to medical practice in Botswana?**

**Sources of information**
▪ Focus group discussion with Phase I Heads of Department, subject heads and block coordinators
▪ Phase I block study guides
▪ Phase II rotation study guides
▪ Phase II (Clinical Years) MBBS curriculum
▪ Morbidity and mortality data for Botswana

**Achievements**
The entire Phase I curriculum is based on a set of PBL cases which have generally been carefully selected to promote important basic science learning in the context of common health care conditions and problems in Botswana. In Phase II, rotations are based on 'cases of the week', almost all of which are clinical cases or conditions commonly present in the country, as can be seen by comparison with national morbidity and mortality data. Both of these steps are correctives to excessive curriculum content.

**Issues to consider/areas for improvement/challenges**
▪ The steps to prevent overload and to achieve relevance have been implicit rather than explicit. There are a few (very few) instances of PBL cases and 'cases of the week' where seemingly undue emphasis is placed on relatively uncommon conditions–although it can be argued that this is done to illustrate concepts such as endocrine feedback loops or the phenomenon of addiction.
▪ As was mentioned in previous questions, there is a need for material especially Phase 2 to be explicitly graded and prioritised, so that not every outcome listed seems to be 'core'.

**Recommendations**
The MBBS Curriculum Committee should instruct the Department of Medical Education to draw up explicit guidelines on preventing overload (after due consultation), and to use these to work with each block and rotation coordinator to edit existing content.

Finally, the Department of Medical Education prepared a review report, which included a brief introduction to the review process, a user-friendly executive summary against the perspective of the MBBS programme, and the findings and recommendations for each question. We submitted the review report and its main findings to the Faculty of Medicine, which accepted it. The FoM tasked its MBBS Curriculum Committee with prioritising recommendations and setting up task teams to implement these progressively.

## Ethical considerations

Data were collected after explaining the nature of the research and obtaining informed consent from each person providing data. We processed data anonymously and stored soft and hard copies of data securely in password-protected files and locked cabinets. Only the research team leader had the documents linking the names of respondents and codes on copies of data, which were similarly stored and protected. A previous research project evaluated graduates' preparedness to function as interns, and these findings were used for some evaluation questions [12]. In June 2017, data collection was concluded.

The authors obtained ethical approval for the research from the following Institutional Review Boards (IRBs): The University of Botswana IRB (# UBR/RES/IRB/1657), the Botswana Government Ministry of Health, Health Research and Development Division (# HPDME:13/18/1 Vol X(479)), Princess Marina Hospital IRB (#PMH 5/79(271-1-2016)) and Mahalapye District Health Management Team IRB (MH/DHMT/1/717(16)).

## Validation

In evaluating the MBBS programme of the University of Botswana, we developed the 'six-step process' described above. In this section, we share our reflection using Cilliers's ten features of complex systems to determine whether the process was adequate in evaluating a complex system: whether it enabled us to see 'the complexity emerging as a result of the patterns of interaction between the elements' [22] more clearly and to formulate useful recommendations accordingly. Cilliers's ten features will be drawn more or less verbatim from his book 'Complexity and Postmodernism: Understanding Complex Systems' [29].

*Cilliers (i): Complex systems consist of a large number of elements.*

The MBBS programme is made up of many elements, e.g. an outcomes-based, competency-based curriculum model. Each research question investigates at least one such element in the programme, so we considered 90 or more of these.

*Cilliers (ii): The elements have to interact, and this interaction must be dynamic. The interactions do not have to be physical; they can also be thought of as the transference of information.*

Of the 90 evaluation questions in the review, 68 dealt with relationships and communication between elements and stakeholders. For Level 2 of the evaluation the extent of such interactions between review questions is shown in (Fig 2).

*Cilliers (iii): The interaction is fairly rich, i.e., any element in the system influences, and is influenced by, quite a few others.*

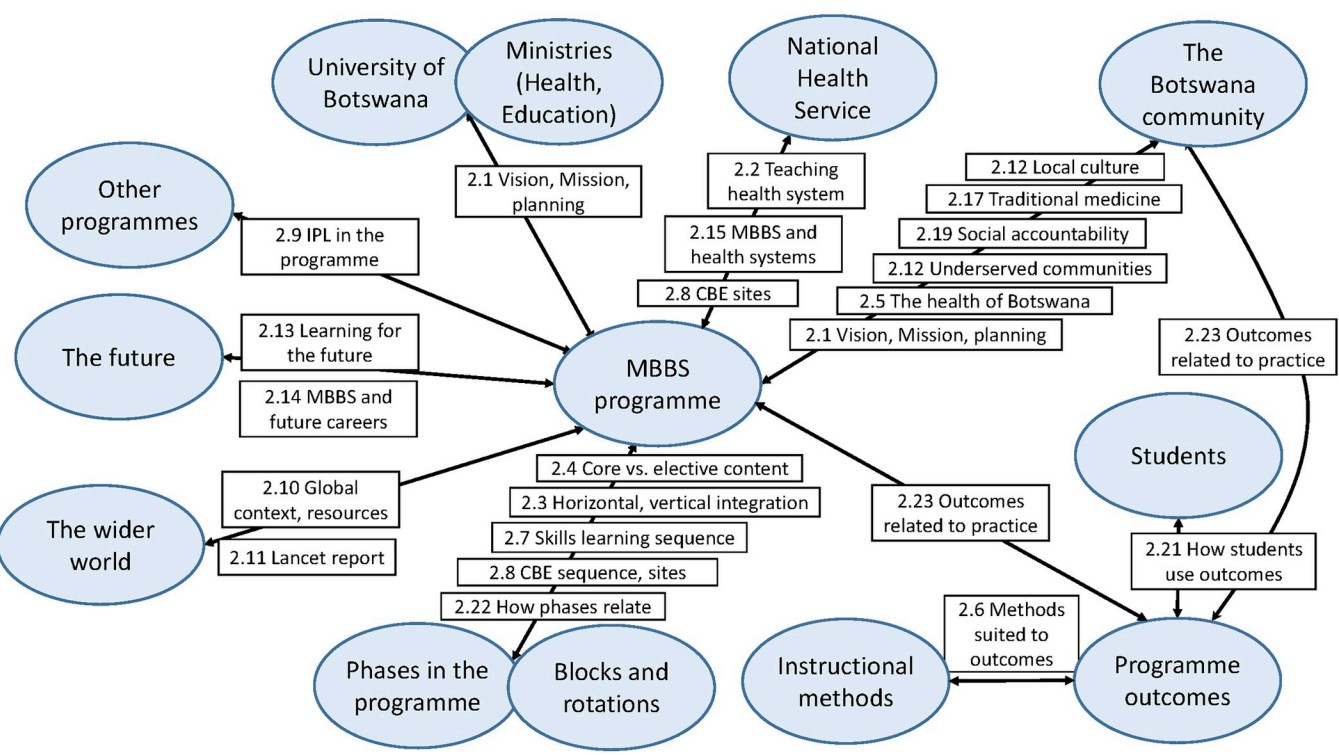

**Fig 2. Evaluation questions addressing relationships and communication in Level 2.**

In the review, a given 'Source of information' could provide data for many research questions (or elements)–for example, an interview with a Head of Department provided data for 24 questions.

*Cilliers (iv): The interactions are non-linear. Non-linearity also guarantees that small causes can have large results and vice versa. It is a precondition for complexity.*

The extensive use of triangulation illustrates our understanding that each element in the MBBS programme is likely to be influenced by more than one cause or factor. For instance, the review showed that the lack of a staffing agreement (a small cause) had a significant effect on the quality of clinical training.

*Cilliers (v): The interactions usually have a fairly short range. This does not preclude wide-ranging influence . . . which is covered in a few steps. . . and the influence gets modulated along the way.*

The review provided ample evidence of immediate or short-range interactions, e.g., between staff and students at the bedside, between exam papers, and the way students study. More wide-ranging influences uncovered included where national health policy and morbidity and mortality data influence curriculum documents and, eventually, actual teaching, assessment, and learning.

*Cilliers (vi): There are loops in the interactions. This feedback can be positive or negative: both kinds are necessary.*

The review uncovered numerous examples of both kinds of feedback. Teaching and learning processes which work well stabilise them in the minds of the teachers involved; those that did not (as shown, for example, by assessment results) were being questioned.

*Cilliers (vii)*: *Complex systems are usually open systems, i.e., they interact with their environment. The scope of the system is usually determined by the purpose of the description of the system and is thus often influenced by the position of the observer observer's position.*

As observers, we framed the MBBS programme as an open system from start to finish. We considered the MBBS system to be interacting with a range of stakeholders: no fewer than 13 questions were framed to explore these interactions.

*Cilliers (viii)*: *Complex systems operate under conditions far from equilibrium. There has to be a constant flow of energy to maintain the organisation of the system and to ensure its survival.*

The review findings reveal a relatively young, unstable system with a number of achievements and many areas which need and are receiving attention and improvement. Even during the nine months of the review, it was apparent that new developments were taking place.

*Cilliers (ix)*: *Complex systems have a history. They evolve through time . . . their past is co-responsible for their present behaviour.*

The review was cognisant of the programme as a system with a history. There were questions focusing on the past (how the programme was planned) and the future (how well the programme will meet projected future needs).

*Cilliers (x)*: *Each element in the system is ignorant of the behaviour of the system as a whole, it responds only to information that is available to it locally. Complexity results from a rich interaction of simple elements that only respond to the limited information each of them is presented with.*

Numerous interactions with numerous programme stakeholders clearly revealed their 'local' perceptions of the processes in which they are involved. As evaluators, we were able to combine these 'local' concepts into a deeper understanding of the overall complexity of the programme.

In summary: complex systems according to Cilliers have ten features. The 'six-step process' illuminated each of these features in the MBBS programme in turn, to varying degrees but omitting none.

## Discussion

In this paper, we contribute to the existing literature by describing a 'six-step process' for comprehensively evaluating an undergraduate medical programme as a complex system. We also reflect on our process using Cilliers's ten features [29], which have been identified as a framework to guide the evaluation of a complex system. We now discuss the lessons we learned while using the 'six-step process.' They are: starting the process with complexity theory at the forefront; being as inclusive as possible in data collection; also applying complexity theory to

the evaluation of smaller programme components as 'mini-complex systems'; and managing one's inevitable insider bias.

## Start the process with complexity theory at the forefront

In our comprehensive review, complexity theory guided our thinking as we selected the combined model we used–but this was only implicit. However, Frye and Hemmer rightly recommend that programme evaluators should identify the theory that underpins the evaluation model they intend to use [22]. 'Complexity' is becoming increasingly popular as an explanatory or theoretical guide [27] in medical education literature, and there are multiple approaches to complexity science to draw from. Cilliers's 'ten features of complex systems' [29] is one of those approaches that can be used for comprehensive programme review.

## Be as inclusive as possible in your data collection

In our experience, investigating many elements (rather than only a few) brought us closer to an 'understanding' of the complex whole, which is consistent with previous literature [25]. It is essential to be as inclusive as possible about the sources of information consulted about each element. Triangulation enables one to understand each element (which could be complex) better. Each situation, process, or outcome observed results from myriad seemingly random events. By triangulation, we can include some of these random events in our understanding, modelling each element as a 'mini-complex system'.

## Apply complexity theory to the evaluation of smaller programme components as 'mini-complex systems'

In addition, we believe that our understanding that each of the 90 elements we investigated is, in fact, a 'mini-complex system' is an important and useful one. It should contribute to an appreciation of other evaluation studies which evaluate limited components of educational programmes, and which also claim to be based on complexity theory [5, 9, 33]. The understanding of 'mini-complex systems' forming part of a greater whole is not clearly spelled out in such studies, and we believe could usefully inform readers. This understanding may also assist other researchers who, in the future, wish to base more limited evaluation studies on complexity theory.

## Manage your inevitable insider bias

Reflecting on ourselves as reviewers of the programme we realise that our being 'insiders' to the MBBS programme brought both advantages and potential disadvantages to the evaluation process. As insiders, we bring a particular understanding of the programme to the evaluation, which outsiders might lack. On the other hand, we also influence what we are evaluating and bring our inevitable biases to the process: as Mezirow points out, we have our fixed 'frames of reference', 'habits of thought', and 'points of view' [45]–an inevitable corollary of the interpretivist approach. We tried to counter our necessarily limited insights and inevitable biases through selecting many elements of the system to investigate; extensive use of triangulation (and thereby enhancing the likely validity of the admittedly inductive reasoning we used in data analysis); conducting data analysis as a team; and explicitly structuring analysis to identify what is working or not, guided by the standards allocated to each element.

## Conclusion and utility

We conclude that the 'six-step process' was valid for evaluating this undergraduate medical programme as a complex system, as defined by Cilliers [29]. The 'six-step process' as it stands

or in adaptation is likely to be useful in similar situations, where evaluators perceive the object of their evaluation to be a complex system. The understanding that elements of complex systems are 'mini-complex systems' in themselves should also contribute to an understanding of studies which evaluate limited components of complex educational programmes.

## Supporting information

**S1 Text. WFME standards.**
(DOCX)

**S2 Text. Other standards.**
(DOCX)

**S3 Text. Questions, data sources, instruments.**
(DOCX)

## Acknowledgments

The authors wish to acknowledge the contributions of the review steering committee in carrying out this demanding project: Tadele Benti, Enoch Sepako, John Wright, Francis Banda, Francesca Cainelli, Oathokwa Nkomazana, Stephane Tshitenge and Julius Mwita.

## Author Contributions

**Conceptualization:** Mpho S. Mogodi, Mmoloki C. Molwantwa.

**Data curation:** Detlef R. Prozesky.

**Formal analysis:** Detlef R. Prozesky, Masego B. Kebaetse, Mpho S. Mogodi, Mmoloki C. Molwantwa.

**Investigation:** Detlef R. Prozesky, Masego B. Kebaetse, Mpho S. Mogodi, Mmoloki C. Molwantwa.

**Methodology:** Detlef R. Prozesky, Mpho S. Mogodi, Mmoloki C. Molwantwa.

**Project administration:** Detlef R. Prozesky.

**Validation:** Detlef R. Prozesky.

**Writing – original draft:** Detlef R. Prozesky, Masego B. Kebaetse, Mpho S. Mogodi, Mmoloki C. Molwantwa.

**Writing – review & editing:** Detlef R. Prozesky, Masego B. Kebaetse, Mpho S. Mogodi, Mmoloki C. Molwantwa.

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
