## [Decision Letter · Decision Letter 0]

30 May 2024

PONE-D-24-08357Developing an evaluation approach for the in-depth review of a new undergraduate medical programme as a complex systemPLOS ONE

Dear Dr. Prozesky,

Thank you for submitting your manuscript to PLOS ONE. After careful consideration, we feel that it has merit but does not fully meet PLOS ONE’s publication criteria as it currently stands. Therefore, we invite you to submit a revised version of the manuscript that addresses the points raised during the review process.

**ACADEMIC EDITOR:**

**you are advised to revise before reassessment.**

Please include the following items when submitting your revised manuscript:A rebuttal letter that responds to each point raised by the academic editor and reviewer(s). You should upload this letter as a separate file labeled 'Response to Reviewers'.A marked-up copy of your manuscript that highlights changes made to the original version. You should upload this as a separate file labeled 'Revised Manuscript with Track Changes'.An unmarked version of your revised paper without tracked changes. You should upload this as a separate file labeled 'Manuscript'.

We look forward to receiving your revised manuscript.

Kind regards,

Ranjit Kumar Dehury

Academic Editor

PLOS ONE

Journal Requirements:

Reviewers' comments:

Reviewer's Responses to Questions

**Comments to the Author**

1. Is the manuscript technically sound, and do the data support the conclusions?

Reviewer #1: Yes

Reviewer #2: Yes

2. Has the statistical analysis been performed appropriately and rigorously? 

Reviewer #1: N/A

Reviewer #2: N/A

3. Have the authors made all data underlying the findings in their manuscript fully available?

Reviewer #1: No

Reviewer #2: No

4. Is the manuscript presented in an intelligible fashion and written in standard English?

Reviewer #1: Yes

Reviewer #2: Yes

5. Review Comments to the Author

Reviewer #1: Background:

• The authors have provided a detailed overview of program evaluation and systems theory. The information provided is beneficial; however, conciseness is needed.

• Several citations are old. For example, starting in line 100, the authors have a whole paragraph discussing the relationships between the elements and their contributions to the system. The key citation in this paragraph (Von Bertalanffy, 1968) is from 1968!

• I would suggest trimming the background to provide readers with concise information supported by recent theories and findings.

• Several citations are outdated. For example, Prigogine, 1997 and Wakeford, 1980. Please provide updated references if possible.

• One suggestion to make the background concise is to provide supporting visual representation instead of text.

• For systems theory, will it be possible to provide some visuals of complex systems? Showing interaction between systems components, interaction with the surroundings etc. will be helpful for readers to grasp the complex idea in a concise manner.

• In the background, provide information on the context of your program development. Why was developing/introducing a new program necessary?

Step1: Setting out the aim

• Line 177: The authors wrote: “The team planning the review unanimously decided that we needed to understand the MBBS programme in its ‘wholeness’ [25]. We realised that..” It is unclear who the planning team is and who the ‘we’ are. Clarify how many authors were part of the planning team and how many of the authors were on the steering committee.

• Who was invited to the workshop? Faculty only? Staff? Please mention them in the text.

Table-1

• In Table-1, the column ‘Originator’ is difficult to comprehend. Consider using ‘Guiding Evaluation Models’ or something similar.

Table-2

• In Table-2, add a column showing which instrument was used to answer these questions. For example, focus groups, surveys, interviews, attendance, student performance etc.

• In the appropriate location in the main text, also mention why one instrument was chosen over another for different questions.

Step 6: Data analysis and presentation

• Line 265: Authors mentioned: “We familiarised ourselves with the data, identified emerging themes, and coded them under two headings: ‘achievements’ and ‘issues to consider/areas for improvement/challenges’.” Where are these results? The way it is presented in table 4 is not sufficient. Please provide the subthemes and representative quotes in a separate table.

• The authors mentioned: “We considered each question as a free-standing unit and brought all the data (qualitative and quantitative) about it together.” Please show these results in graphs. Put the survey questionnaire in the supporting materials.

• Provide a flowchart or diagram outlining the timeline and the major steps of this program evaluation.

General Minor Edits:

• Line 22: As you have removed the term ‘new’ from the title, please remove ‘new’ from the abstract as well.

• Punctuation (.) is missing in line 77.

• Line 111: processes can written as ‘processes’ to match with ‘things.’

• Line 130: rewrite the sentence for clarity. A suggestion: The reported applications of the theory included making conceptual frameworks and planning data analysis and interpretation.

• In paragraph starting at line 156: Use passive voice. Suggestion: ‘They elected a Steering Committee’ can be A steering committee was elected.

• Line 420: should be ‘spelled’

• Reference # 3, 25 and 32 are too old. Please replace with updated literature.

• Update supporting information section with the list of all documents: Information about the ethics approval letters are missing in this section.

Reviewer #2: The manuscript is useful for programme evaluation research. It provides a well-defined framework to measure the effectiveness of an academic curriculum. A detailed step-by-step process is explained. The background is supported by sufficient literature.

The arguments are supported by the data. Data are available on demand and supporting documents are provided as supplementary material.

The authors have complied with the previous comments and have strengthened the paper. It can be accepted for publication.

6. PLOS authors have the option to publish the peer review history of their article (what does this mean?). If published, this will include your full peer review and any attached files.

Reviewer #1: No

Reviewer #2: No

---

## [Author Response · Author response to Decision Letter 0]

19 Aug 2024

Please note that the line numbers below refer to the clean copy of the manuscript. We have also highlighted the required changes in the clean copy.

No. Reviewers’ Comments Authors’ feedback

 General Comments

1. Is the manuscript technically sound, and do the data support the conclusions?

Reviewer #1: Yes

Reviewer #2: Yes 

We thank the reviewer for this encouraging comment.

2. Has the statistical analysis been performed appropriately and rigorously?

Reviewer #1: N/A

Reviewer #2: N/A 

We thank the reviewer for this encouraging comment.

3. Have the authors made all data underlying the findings in their manuscript fully available?

Reviewer #1: No

Reviewer #2: No 

Thank you for this suggestion. In this case there are restrictions in place that prevent the public sharing of our minimal data. The consent forms we used asked study participants to agree to ‘provide the information requested, which will be incorporated (anonymously) in the Review Report only’. We are therefore not able to make these data available without restrictions. 

We are following the alternative route you recommend, as follows:

Readers which wish to have access to the data supporting this research must please apply to: The Office for Research and Development, University of Botswana (research.ethics@ub.ac.bw) 

4. Is the manuscript presented in an intelligible fashion and written in standard English?

PLOS ONE does not copy edit accepted manuscripts, so the language in submitted articles must be clear, correct, and unambiguous. Any typographical or grammatical errors should be corrected at revision, so please note any specific errors here.

Reviewer #1: Yes

Reviewer #2: Yes 

We thank the reviewer for this encouraging comment.

5. Reviewer #1: Background:

• The authors have provided a detailed overview of program evaluation and systems theory. The information provided is beneficial; however, conciseness is needed.

• I would suggest trimming the background to provide readers with concise information supported by recent theories and findings.

• One suggestion to make the background concise is to provide supporting visual representation instead of text.

• For systems theory, will it be possible to provide some visuals of complex systems? Showing interaction between systems components, interaction with the surroundings etc. will be helpful for readers to grasp the complex idea in a concise manner. We thank the reviewer for these comments. 

In response to the previous round of feedback we revised the background to make it more succinct and focused – we in fact trimmed it from 5½ pages (1413 words) 3½ pages (894 words) – this was in spite of also having moved discussion of Cilliers’s model from a later section of the document to the Background. We also structured the Background more clearly around the following: reasons for program evaluation (66-71), scope of evaluation (71-75), data collection methods used (75-80), standards of evaluation used (82-87), and frequently applied theoretical frameworks and models applied in programme evaluation (89-121). This last section is more detailed because the issue of theory is at the heart of this work and needed to be explored thoroughly.

We are concerned that further trimming of the introduction may compromise our overall argument. In the literature review we sought to provide evidence that our evaluation is unique in terms of its combination of scope, methods and theoretical grounding.

6. • Several citations are old. For example, starting in line 100, the authors have a whole paragraph discussing the relationships between the elements and their contributions to the system. The key citation in this paragraph (Von Bertalanffy, 1968) is from 1968!

• Several citations are outdated. For example, Prigogine, 1997 and Wakeford, 1980. Please provide updated references if possible.

Reference # 3, 25 and 32 are too old. Please replace with updated literature. 

We thank the reviewer for this comment. 

References 3 has been changed to a similar more recent one.

References 25, 29, 30, 32 and 45 are from seminal original literature referring to models, theories and frameworks that form an important part of the discussion – Prigogine, Cilliers, Von Bertlanffy, Stufflebeam, Mezirow are all internationally respected as initiators of valuable concepts and methods

7. In the background, provide information on the context of your program development. Why was developing/introducing a new program necessary? 

We thank the reviewer for this comment. 

We have inserted a statement of the rationale for this programme under ‘Context’ – line 149-151

8. Line 177: The authors wrote: “The team planning the review unanimously decided that we needed to understand the MBBS programme in its ‘wholeness’ [25]. We realised that..” It is unclear who the planning team is and who the ‘we’ are. Clarify how many authors were part of the planning team and how many of the authors were on the steering committee.

• Who was invited to the workshop? Faculty only? Staff? Please mention them in the text.

We thank the reviewer for this comment. 

The position of the authors at the different stages of the evaluation has been clarified in lines 165-166. 

Attendance at the workshop is set out in lines 161-162 and 163-164.

9. • In Table-1, the column ‘Originator’ is difficult to comprehend. Consider using ‘Guiding Evaluation Models’ or something similar. 

We thank the reviewer for this comment. The correction from ‘Originator’ to ‘Guiding Evaluation Models’ in Table 1 has been made (line 200).

10. • In Table-2, add a column showing which instrument was used to answer these questions. For example, focus groups, surveys, interviews, attendance, student performance etc.

• In the appropriate location in the main text, also mention why one instrument was chosen over another for different questions. 

We thank the reviewer for this comment. The column ‘Instruments used to collect data’ has been added to Table 2 (line 234).

11. • Line 265: Authors mentioned: “We familiarised ourselves with the data, identified emerging themes, and coded them under two headings: ‘achievements’ and ‘issues to consider/areas for improvement/challenges’.” Where are these results? The way it is presented in table 4 is not sufficient. Please provide the subthemes and representative quotes in a separate table. 

We thank the reviewer for this comment. In this research we did not intend to conduct inductive qualitative analysis. The pragmatic process we used is the following:

We noted the question and its number.

Using a prepared guide which listed the instruments providing data for each question, we located all the data provided for that question

We considered all of these data together and looked for recurring patterns, enabling us to list ‘achievements’ and ‘issues to consider/ areas for improvement/ challenges’ related to that question. 

The process we followed therefore did not employ ‘subthemes’ and ‘representative quotes’. 

12. • The authors mentioned: “We considered each question as a free-standing unit and brought all the data (qualitative and quantitative) about it together.” Please show these results in graphs. Put the survey questionnaire in the supporting materials.

• Provide a flowchart or diagram outlining the timeline and the major steps of this program evaluation.

We thank the reviewer for this comment. However the focus of the paper was not to demonstrate the rigour of the data analysis (which we believe was rigorous, as explained above) but rather to demonstrate the approach followed – triangulating multiple sources of data as a way of viewing our particular complex system appropriately. 

As explained above the results were recorded narratively and not graphically.

The research used 96 instruments most of which provided data for several questions – only 10 of these were questionnaires. Most instruments were used several times and between them provided 320 separate pieces of data. For pragmatic reasons it is not feasible to append all of these as supporting materials.

A diagram has been provided at line 179. 

13. Line 22: As you have removed the term ‘new’ from the title, please remove ‘new’ from the abstract as well. 

We thank the reviewer for this comment. The change has been made.

14. Punctuation (.) is missing in line 77. 

We thank the reviewer for this comment. This has been corrected. 

15. Line 111: processes can written as ‘processes’ to match with ‘things.’

We thank the reviewer for this comment. Complexity theory makes a particular distinction between ‘processes’ and ‘things’. To make this change would modify the sense of complexity theory.

16. Line 130: rewrite the sentence for clarity. A suggestion: The reported applications of the theory included making conceptual frameworks and planning data analysis and interpretation. 

We thank the reviewer for this comment. This change has been made.

17. In paragraph starting at line 156: Use passive voice. Suggestion: ‘They elected a Steering Committee’ can be A steering committee was elected. 

We thank the reviewer for this comment. This change has been made.

18. Line 420: should be ‘spelled’ 

We thank the reviewer for this comment. We have made the change. 

19. Update supporting information section with the list of all documents: Information about the ethics approval letters are missing in this section. 

We thank the reviewer for this comment.

Under point 12 above we mentioned that there are 320 documents containing data. We respectfully suggest that the information needed to grasp the scope of the review is given in Table 3, so a list of all the documents would be redundant.

Information about the ethics approval letters is given in lines 298-302.

---

## [Decision Letter · Decision Letter 1]

14 Oct 2024

Developing an evaluation approach for the in-depth review of a new undergraduate medical programme as a complex system

PONE-D-24-08357R1

Dear Dr. Prozesky,

We’re pleased to inform you that your manuscript has been judged scientifically suitable for publication and will be formally accepted for publication once it meets all outstanding technical requirements.

Kind regards,

Ranjit Kumar Dehury

Academic Editor

PLOS ONE

Additional Editor Comments (optional):

The article is of publishable standard and accepted.

Reviewers' comments:

Reviewer's Responses to Questions

**Comments to the Author**

1. If the authors have adequately addressed your comments raised in a previous round of review and you feel that this manuscript is now acceptable for publication, you may indicate that here to bypass the “Comments to the Author” section, enter your conflict of interest statement in the “Confidential to Editor” section, and submit your "Accept" recommendation.

Reviewer #1: All comments have been addressed

Reviewer #2: All comments have been addressed

2. Is the manuscript technically sound, and do the data support the conclusions?

Reviewer #1: Yes

Reviewer #2: Yes

3. Has the statistical analysis been performed appropriately and rigorously? 

Reviewer #1: N/A

Reviewer #2: N/A

4. Have the authors made all data underlying the findings in their manuscript fully available?

Reviewer #1: No

Reviewer #2: No

5. Is the manuscript presented in an intelligible fashion and written in standard English?

Reviewer #1: Yes

Reviewer #2: Yes

6. Review Comments to the Author

Reviewer #1: The manuscript has been sufficiently revised and improved. The authors have responded to most of the comments. Further improvements can be made by providing details of the qualitative data analysis.

Reviewer #2: (No Response)

7. PLOS authors have the option to publish the peer review history of their article (what does this mean?). If published, this will include your full peer review and any attached files.

Reviewer #1: No

Reviewer #2: **Yes: **Imteyaz Ahmad

---

## [Editor Report · Acceptance letter]

10 Dec 2024

PONE-D-24-08357R1 

PLOS ONE

Dear Dr. Prozesky, 

I'm pleased to inform you that your manuscript has been deemed suitable for publication in PLOS ONE. Congratulations! Your manuscript is now being handed over to our production team.

Kind regards, 

on behalf of

Dr. Ranjit Kumar Dehury 

Academic Editor

PLOS ONE